# Noisy $\ell^0$-Sparse Subspace Clustering on Dimensionality Reduced Data

**Yingzhen Yang**

School of Computing and Augmented Intelligence
Arizona State University
699 S Mill Ave. Tempe, AZ 85281, USA
`yingzhen.yang@asu.edu`

**Ping Li**

Cognitive Computing Lab
Baidu Research
10900 NE 8th ST. Bellevue, WA 98004, USA
`pingli98@gmail.com`

## Abstract

[a]Sparse subspace clustering methods with sparsity induced by $\ell^0$-norm, such as $\ell^0$-Sparse Subspace Clustering ($\ell^0$-SSC) [Yang et al., 2018], are demonstrated to be more effective than its $\ell^1$ counterpart such as Sparse Subspace Clustering (SSC) [Elhamifar and Vidal, 2013]. However, the theoretical analysis of $\ell^0$-SSC is restricted to clean data that lie exactly in subspaces. Real data often suffer from noise and they may lie close to subspaces. In this paper, we show that an optimal solution to the optimization problem of noisy $\ell^0$-SSC achieves subspace detection property (SDP), a key element with which data from different subspaces are separated, under deterministic and semi-random model. Our results provide theoretical guarantee on the correctness of noisy $\ell^0$-SSC in terms of SDP on noisy data for the first time, which reveals the advantage of noisy $\ell^0$-SSC in terms of much less restrictive condition on subspace affinity. In order to improve the efficiency of noisy $\ell^0$-SSC, we propose Noisy-DR-$\ell^0$-SSC which provably recovers the subspaces on dimensionality reduced data. Noisy-DR-$\ell^0$-SSC first projects the data onto a lower dimensional space by random projection, then performs noisy $\ell^0$-SSC on the projected data for improved efficiency. Experimental results demonstrate the effectiveness of Noisy-DR-$\ell^0$-SSC.

[a]Yingzhen Yang's work was conducted as a consulting researcher at Baidu Research - Bellevue, WA, USA.

## 1 INTRODUCTION

Clustering is an important unsupervised learning procedure for analyzing a broad class of scientific data. High-dimensional data, such as facial images and gene expression data, often lie in low-dimensional subspaces in many cases, and clustering in accordance to the underlying subspace structure is particularly important. Among various subspace clustering algorithms, the ones that employ sparsity prior, such as Sparse Subspace Clustering (SSC) [Elhamifar and Vidal, 2013] and $\ell^0$-Sparse Subspace Clustering ($\ell^0$-SSC) [Yang et al., 2018], have been proven to be effective in separating the data in accordance with the subspaces that the data lie in under certain assumptions. Furthermore, Sparse Additive Subspace Clustering [Yuan and Li, 2014] considers a nonlinear transformation of each data point such that the transformed point can be linearly represented by data in the same subspace as that point, extending the usual linear representation by SSC.

Sparse subspace clustering methods construct the sparse similarity matrix by sparse representation of the data. Subspace detection property (SDP) defined in Section 2 ensures that the similarity between data from different subspaces vanishes in the sparse similarity matrix, and applying spectral clustering [Ng et al., 2001] on such sparse similarity matrix leads to compelling clustering performance. Elhamifar and Vidal [Elhamifar and Vidal, 2013] prove that when the subspaces are independent or disjoint, SDP can be satisuyfied by solving the canonical sparse linear representation problem using data as the dictionary, under certain conditions on the rank, or singular value of the data matrix and the principle angle between the subspaces. Under the independence assumption on the subspaces, low rank representation [Liu et al., 2013, Liu and Li, 2014, 2016] is also proposed to recover the subspace structures. Relaxing the assumptions on the subspaces to allowing overlapping subspaces, the Greedy Subspace Clustering [Park et al., 2014] and the Low-Rank Sparse Subspace Clustering [Wang et al., 2013] achieve subspace detection property with high probability. The geometric analysis in Soltanolkotabi and Candés [2012] shows the theoretical results on subspace recovery by SSC. In the following, we use the term SSC or $\ell^1$-SSC exchangeably to indicate the Sparse Subspace Clustering method in Elhamifar and Vidal [2013].

Real data often suffer from noise. The correctness of noisy SSC is analyzed in Wang and Xu [2013] which handles noisy data that lie close to disjoint or overlapping subspaces,

*Accepted for the 38th Conference on Uncertainty in Artificial Intelligence* (UAI 2022).

Table 1: Comparison between Different Subspace Clustering Methods in terms of Conditions Required under the Semi-Random Model. Please refer to Section 1.2 for the definition of notations.

| Methods or Assumptions | Allowing Overlapping Subspaces | Subspace Affinity |
|---|---|---|
| Greedy Subspace Clustering (GSC) [Park et al., 2014] | Yes | $\max_{k,l \in [K]} \mathrm{aff}(\mathcal{S}_k, S_l) < \frac{C_2 \log n / K}{\log(d_0 L \delta^{-1}) \cdot \log(n d_0 \delta^{-1})} \overset{d_0 \to \infty}{\longrightarrow} 0$ |
| $\ell^1$-SSC [Elhamifar and Vidal, 2011], Noisy SSC [Wang and Xu, 2013] | Yes | $\max_{k,l \in [K]} \mathrm{aff}(\mathcal{S}_k, S_l) < \sqrt{d_0} \cdot \frac{\bar{c}\sqrt{\log \rho}}{8\sqrt{2}\log n} \overset{d_0 \to \infty}{\longrightarrow} 0$ |
| Affine Sparse Subspace Clustering (ASSC) [Li et al., 2018, You et al., 2019] | No | NA |
| Noisy $\ell^0$-SSC [Yang et al., 2018] | Yes | $\max_{k,l \in [K]} \mathrm{aff}(\mathcal{S}_k, S_l) < \frac{\sigma_{\min}'^2}{\tau_0 - 1} > 0$ for sufficiently large $d_0$ |

and the original optimization problem of noisy SSC is proposed in Elhamifar and Vidal [2013]. While Yang et al. [2016], Yang [2018] prove the correctness of $\ell^0$-SSC or its dimensionality reduced variant on clean data based on a constrained $\ell^0$-minimization problem, it empirically solves an unconstrained $\ell^0$-regularized problem to handle noise in data, and they lack theoretical analysis on the correctness of $\ell^0$-SSC on noisy data. fhandles noisy data that lie close to disjoint or overlapping subspaces. While $\ell^0$-SSC [Yang et al., 2018] has guaranteed clustering correctness via subspace detection property under much milder assumptions than previous subspace clustering methods including SSC, it assumes that the observed data lie in exactly in the subspaces and does not handle noisy data. In this paper, we present noisy $\ell^0$-SSC, which enhances $\ell^0$-SSC by theoretical guarantee on the correctness of clustering on noisy data. It should be emphasized that while $\ell^0$-SSC on clean data [Yang et al., 2018] empirically adopts a form of optimization problem robust to noise, it lacks theoretical analysis on the correctness of $\ell^0$-SSC on noisy data. In this paper, the correctness of noisy $\ell^0$-SSC on noisy data in terms of the subspace detection property is established. Our analysis is under both deterministic model and semi-random model, which are the models employed in the geometric analysis of SSC [Soltanolkotabi and Candés, 2012]. Our randomized analysis demonstrates the advantage of noisy $\ell^0$-SSC over its $\ell^1$ counterpart as more general assumption on data distribution can be adopted. Moreover, we present Noisy Dimensionality Reduced $\ell^0$-Sparse Subspace Clustering (Noisy-DR-$\ell^0$-SSC), an efficient version of noisy $\ell^0$-SSC which also enjoys robustness to noise. Noisy-DR-$\ell^0$-SSC first projects the data onto a lower dimensional space by random projection, then performs noisy $\ell^0$-SSC on the dimensionality reduced data. Noisy-DR-$\ell^0$-SSC provably recovers the underlying subspace structure in the original data from the projected data under deterministic model. Experimental results show the effectiveness of both noisy $\ell^0$-SSC and Noisy-DR-$\ell^0$-SSC.

## 1.1 NOTATIONS

We use bold letters for matrices and vectors, and regular lower letter for scalars throughout this paper. The bold letter with superscript indicates the corresponding column of a matrix, e.g. $\mathbf{A}^i$ is the $i$-th column of matrix $\mathbf{A}$, and the bold letter with subscript indicates the corresponding element of a matrix or vector. $\|\cdot\|_F$ and $\|\cdot\|_p$ denote the Frobenius

norm and the vector $\ell^p$-norm or the matrix $p$-norm, and $\|\cdot\|_0$ is the $\ell^0$-norm, that is, the number of nonzero elements of a vector. $\mathrm{diag}(\cdot)$ indicates the diagonal elements of a matrix. $\mathbf{H_T} \subseteq \mathbb{R}^d$ indicates the subspace spanned by the columns of $\mathbf{T}$, and $\mathbf{A_I}$ denotes a submatrix of $\mathbf{A}$ whose columns correspond to the nonzero elements of $\mathbf{I}$ (or with indices in $\mathbf{I}$ without confusion). $\sigma_t(\cdot)$ denotes the $t$-th largest singular value of a matrix, and $\sigma_{\min}(\cdot)$ indicates the smallest singular value of a matrix. $\mathrm{supp}(\cdot)$ is the support of a vector, $\mathbb{P}_{\mathcal{S}'}$ is the operator of orthogonal projection onto the subspace $\mathcal{S}'$. $[n]$ represents all the natural numbers between 1 and $n$ inclusively. $\mathbb{S}^{d-1}$ denotes the unit sphere in $\mathbb{R}^d$. $\Theta(a)$ denotes a number such that there exists two constants $c_1$ and $c_2$ such that $\Theta(a) \in [c_1 a, c_2 a]$.

## 1.2 CONTRIBUTIONS

First, the correctness of noisy $\ell^0$-SSC on noisy data in terms of the subspace detection property is established for the first time, which is presented in Section 3 of this paper. Our analysis is under both deterministic model and semi-random model, which are also the models employed by the geometric analysis of SSC [Soltanolkotabi2012]. Our randomized analysis demonstrates the significant advantage of noisy $\ell^0$-SSC over its $\ell^1$ counterpart and other competing subspace clustering methods in terms of much less restrictive condition on the subspace affinity. Table 1 below demonstrates the conditions under which SDP holds for representative subspace clustering methods under the semi-random model, including Greedy Subspace Clustering (GSC) [Park et al., 2014], $\ell^1$-SSC [Elhamifar and Vidal, 2011], Noisy SSC [Wang and Xu, 2013], Affine Sparse Subspace Clustering (ASSC) [Li et al., 2018, You et al., 2019], Noisy $\ell^0$-SSC [Yang et al., 2018]. When the size of data $n$ grows exponentially in terms of the common subspace dimension $d_0$ (the dimension of every subspace is $d_0$), in particular, $n = \Theta(e^{d_0^\tau})$ for $\tau \in (0.5, 0.9)$, then all the competing subspace clustering methods other than Noisy $\ell^0$-SSC either do not allow overlapping subspaces, or require the maximum pairwise subspace affinity goes to 0 when $d_0 \to \infty$, which means that these methods require all the subspaces to be almost pairwise orthogonal when the common subspace dimension $d_0$ is very large. Instead, Noisy $\ell^0$-SSC allows subspace affinity to be lower bounded from 0, suggesting that Noisy $\ell^0$-SSC is still able to recover subspaces which are not orthogonal when $d_0$ is very large. $\sigma_{\min}'$ in Table 1 is defined in Theorem 3.6.

In Table 1, it is preferred that a subspace clustering method requires milder conditions, which are allowing overlapping subspaces and larger upper bound for the maximum subspace affinity, denoted by $\max_{k,l \in [K]} \text{aff}(\mathcal{S}_k, S_l)$ where $\{\mathcal{S}_k\}_{k=1}^K$ are $K$ subspaces, so that the underlying subspaces can be recovered for overlapping subspaces and for subspaces which are closer to each other (larger subspace affinity). Here aff denotes subspace affinity, $d_0$ is the common subspace dimension, $\delta$ is a small positive constant (see [Park et al., 2014]). $r_0 > 1$ is an upper bound for the support of an optimal solution to the noisy $\ell^0$-SSC problem for all data points. In this table, $n = \Theta(e^{d_0^\tau})$ for $\tau \in (0.5, 0.9)$ when $d_0 \to \infty$. Note that two subspaces are overlapping subspaces if the dimension of their intersection is larger than 1. When a subspace clustering method does not allow overlapping subspaces, then no condition on subspace affinity is presented in the subspace clustering literature.

Second, we propose Noisy Dimensionality Reduced $\ell^0$-Sparse Subspace Clustering (Noisy-DR-$\ell^0$-SSC) to accelerate noisy $\ell^0$-SSC with provable robustness to noise. Noisy-DR-$\ell^0$-SSC first projects the data onto a lower dimensional space by random projection, then performs noisy $\ell^0$-SSC on the dimensionality reduced data. Two types of random projections are used in Noisy-DR-$\ell^0$-SSC, which are the random projection induced by randomized low-rank approximation and the sparse random projection, particular "Count-Sketch (SC) Projections".

It should be emphasized that Yang [2018] also studies dimensionality reduced $\ell^0$-SSC. However, the analysis of this work is performed on the following constrained $\ell^0$-minimization problem and only for clean data without noise, On the other hand, the actual optimization problem [Yang, 2018] solves a unconstrained $\ell^0$-regularized problem. In contrast, we analyze noisy $\ell^0$-SSC on the unconstrained $\ell^0$-regularized problem with noisy data which reveals the advantage of noisy $\ell^0$ SSC over $\ell^1$-SSC. Our analysis also suggests that a larger $\lambda$ tends to guarantee the subspace detection property (Remark 3.5), verified by experiments. Throughout the paper, we refer to $\ell^0$-SSC for noisy data with the unconstrained $\ell^0$-regularized problem as noisy $\ell^0$-SSC.

## 2  PROBLEM SETUP

Sparse Subspace Clustering (SSC) methods, such as [Elhamifar and Vidal, 2011, Soltanolkotabi and Candés, 2012, Wang and Xu, 2013, Yuan and Li, 2014], construct a sparse similarity matrix by sparse representation of the data, and then perform clustering on the sparse similarity matrix.

We hereby introduce the notations for subspace clustering on noisy data considered in this paper. The uncorrupted data matrix is denoted by $\mathbf{Y} = [\mathbf{y}_1, \ldots, \mathbf{y}_n] \in \mathbb{R}^{d \times n}$, where $d$ is the dimensionality and $n$ is the size of the data. The uncorrupted data $\mathbf{Y}$ lie in a union of $K$ distinct

subspaces $\{\mathcal{S}_k\}_{k=1}^K$ of dimensions $\{d_k\}_{k=1}^K$ with $d_{\max} := \max_{k \in [K]} d_k$ and $d_{\min} := \min_{k \in [K]} d_k$. The observed noisy data is $\mathbf{X} = \mathbf{Y} + \mathbf{N}$, where $\mathbf{N} = [\mathbf{n}_1, \ldots, \mathbf{n}_n] \in \mathbb{R}^{d \times n}$ is the additive noise. $\mathbf{x}_i = \mathbf{y}_i + \mathbf{n}_i$ is the noisy data point that is corrupted by the noise $\mathbf{n}_i$. We let $\mathbf{Y}^{(k)} \in \mathbb{R}^{d \times n_k}$ denote the data belonging to subspace $\mathcal{S}_k$ with $\sum_{k=1}^K n_k = n$, and denote the corresponding columns in $\mathbf{X}$ by $\mathbf{X}^{(k)}$. Let $\mathbf{U}^{(k)} \in \mathbb{R}^{d \times d_k}$ be the orthogonal basis of $\mathcal{S}_k$ for all $k \in [K]$. The data $\mathbf{X}$ are normalized such that each column has unit $\ell^2$-norm in our deterministic analysis. We consider deterministic noise model where the noise $\mathbf{N}$ is fixed and $\max_{i \in [n]} \|\mathbf{n}_i\|_2 \leq \delta$.

Formally, given observed data $\mathbf{X} \in \mathbb{R}^{d \times n}, \mathbf{X} = [\mathbf{x}_1, \ldots, \mathbf{x}_n]$, where $\mathbf{x}_i \in \mathbb{R}^d$, SSC solves the following optimization problem for each $i \in [n]$:

$$\min_{\boldsymbol{\beta} \in \mathbb{R}^n} \|\boldsymbol{\beta}\|_1 \quad \text{s.t. } \mathbf{x}_i = \mathbf{X}\boldsymbol{\beta}, \boldsymbol{\beta}_i = 0. \quad (1)$$

In order to handle noisy data, noisy SSC [Wang and Xu, 2013] was proposed to solve the $\ell^1$ regularized problem:

$$\min_{\boldsymbol{\beta} \in \mathbb{R}^n} \|\mathbf{x}_i - \mathbf{X}\boldsymbol{\beta}\|^2 + \lambda\|\boldsymbol{\beta}\|_1, \quad \text{s.t. } \boldsymbol{\beta}_i = 0. \quad (2)$$

The sparse code $\boldsymbol{\beta}$ of the data point $\mathbf{x}_i$ is obtained by solving (1) or (2) for SSC or noisy SSC. A coefficient matrix $\mathbf{Z} \in \mathbb{R}^{n \times n}$ is then formed by concatenating the sparse codes of all the data points, and the $i$-th column of $\mathbf{Z}$ is the sparse code of $\mathbf{x}_i$. The sparse similarity matrix is then computed by $\mathbf{W} = \frac{|\mathbf{Z}| + |\mathbf{Z}^\top|}{2}$. Subspace detection property (SDP, formally defined later) ensures that the similarity between data from different subspaces vanishes in the sparse similarity matrix. If SDP holds, then similarity between data points from different clusters vanish in $\mathbf{W}$. As a result, performing spectral clustering on $\mathbf{W}$ leads to compelling clustering results.

Under the independence assumption on the subspaces, low rank representation [Liu et al., 2013] is proposed to recover the subspace structures. Relaxing the assumptions on the subspaces to allowing overlapping subspaces, the Greedy Subspace Clustering [Park et al., 2014] and the Low-Rank Sparse Subspace Clustering [Liu et al., 2013] achieve subspace detection property with high probability. The geometric analysis in Soltanolkotabi and Candés [2012] shows the theoretical results on subspace recovery by SSC. In the following text, we use SSC or $\ell^1$-SSC exchangeably to indicate the Sparse Subspace Clustering method in Soltanolkotabi and Candés [2012], Elhamifar and Vidal [2013].

$\ell^0$-SSC [Yang et al., 2018] proposes to solve the following $\ell^0$ sparse representation problem

$$\min_{\mathbf{Z} \in \mathbb{R}^{n \times n}} \|\mathbf{Z}\|_0 \quad s.t. \mathbf{X} = \mathbf{XZ}, \text{ diag}(\mathbf{Z}) = \mathbf{0}, \quad (3)$$

and it proves that SDP is satisfied with an globally optimal solution to problem (3). In Yang et al. [2018], the $\ell^0$ regularized sparse approximation problem below is solved so as

to handle noisy data for $\ell^0$-SSC, which is the optimization problem of noisy $\ell^0$-SSC:

$$\min_{\mathbf{Z} \in \mathbb{R}^{n \times n}, \text{diag}(\mathbf{Z})=0} L(\mathbf{Z}) = \|\mathbf{X} - \mathbf{X}\mathbf{Z}\|_F^2 + \lambda\|\mathbf{Z}\|_0, . \quad (4)$$

The optimization problem of noisy $\ell^0$-SSC (4) is separable. For each $i \in [n]$, the optimization problem with respect to the sparse code $\boldsymbol{\beta}$ of $i$-th data point is

$$\min_{\boldsymbol{\beta} \in \mathbb{R}^n, \boldsymbol{\beta}_i=0} L(\boldsymbol{\beta}) = \|\mathbf{x}_i - \mathbf{X}\boldsymbol{\beta}\|_2^2 + \lambda\|\boldsymbol{\beta}\|_0. \quad (5)$$

The sparse similarity matrix $\mathbf{W}$ is then computed in the same way as $\ell^1$-SSC by $\mathbf{W} = \frac{|\mathbf{Z}|+|\mathbf{Z}^\top|}{2}$, and the subspace clustering result of noisy $\ell^0$-SSC is achieved by performing spectral clustering on $\mathbf{W}$.

In the following text, we always use $\boldsymbol{\beta}^*$ to denote an optimal solution to (5), and define $r^* := \|\boldsymbol{\beta}^*\|_0$.

The definition of subspace detection property for noisy $\ell^0$-SSC and noiseless $\ell^0$-SSC, i.e. $\ell^0$-SSC on noiseless data, is defined in Definition 2.1 below.

**Definition 2.1.** (Subspace detection property for noisy and noiseless $\ell^0$-SSC) Let $\mathbf{Z}^*$ be an optimal solution to (4). The subspaces $\{\mathcal{S}_k\}_{k=1}^K$ and the data $\mathbf{X}$ satisfy the Subspace Detection Property (SDP) for noisy $\ell^0$-SSC if $\mathbf{Z}^i$ is a nonzero vector, and nonzero elements of $\mathbf{Z}^i$ correspond to the columns of $\mathbf{X}$ from the same subspace as $\mathbf{y}_i$ for all $1 \le i \le n$. We say that SDP holds for $\mathbf{x}_i$ if nonzero elements of $\mathbf{Z}^{*i}$, which is $\boldsymbol{\beta}^*$ for problem (5), correspond to the data that lie in the same subspace as $\mathbf{y}_i$, for either noisy $\ell^0$-SSC or noiseless $\ell^0$-SSC.

# 3 ANALYSIS FOR NOISY $\ell^0$-SSC

Similar to Soltanolkotabi and Candés [2012], we introduce the deterministic and the semi-random model for the analysis of noisy $\ell^0$-SSC.

- **Deterministic Model:** the subspaces and the data in each subspace are fixed.

- **Semi-Random Model:** the subspaces are fixed but the data are independent and identically distributed in each of the subspaces.

The data in the above definitions refer to clean data without noise. Both the deterministic model and the semi-random model are extensively employed to analyze the subspace detection property in the subspace learning literature [Soltanolkotabi and Candés, 2012, Wang et al., 2013, Wang and Xu, 2013, Acharyya and Ghosh, 2015].

## 3.1 NOISY $\ell^0$-SSC: DETERMINISTIC ANALYSIS

We first introduce the definition of general position and external subspace before our analysis on noisy $\ell^0$-SSC.

**Definition 3.1.** (General position) For any $1 \le k \le K$, the data $\mathbf{Y}^{(k)}$ are in general position if any subset of $L \le d_k$ data points (columns) of $\mathbf{Y}^{(k)}$ are linearly independent. $\mathbf{Y}$ are in general position if $\mathbf{Y}^{(k)}$ are in general position for $1 \le k \le K$.

The assumption of general condition is rather mild. In fact, if the data points in $\mathbf{X}^{(k)}$ are independently distributed according to any continuous distribution, then they almost surely in general position.

Let the distance between a point $\mathbf{x} \in \mathbb{R}^d$ and a subspace $\mathcal{S} \subseteq \mathbb{R}^d$ be defined as $d(\mathbf{x}, \mathcal{S}) = \inf_{\mathbf{y} \in \mathcal{S}} \|\mathbf{x} - \mathbf{y}\|_2$, the definition of external subspaces is presented as follows.

**Definition 3.2.** (External subspace of limited dimension) For a point $\mathbf{y} \in \mathbf{Y}^{(k)}$, a subspace $\mathbf{H}_{\{\mathbf{y}_{i_j}\}_{j=1}^L}$ spanned by a set of linear independent points $\{\mathbf{y}_{i_j}\}_{j=1}^L \subseteq \mathbf{Y}$ is defined to be an external subspace of $\mathbf{y}$ if $\{\mathbf{y}_{i_j}\}_{j=1}^L \not\subseteq \mathbf{Y}^{(k)}$ and $\mathbf{y} \notin \{\mathbf{y}_{i_j}\}_{j=1}^L$. The set of all external subspaces of $\mathbf{y}$ of dimension no greater than $r$ with $r \ge 1$ for $\mathbf{y}$ is denoted by $\mathcal{H}_{\mathbf{y},r}$, that is, $\mathcal{H}_{\mathbf{y},r} = \{\mathbf{H}: \mathbf{H} = \mathbf{H}_{\{\mathbf{y}_{i_j}\}_{j=1}^L}, \dim[\mathbf{H}] = L, L \le r, \{\mathbf{y}_{i_j}\}_{j=1}^L \not\subseteq \mathbf{Y}^{(k)}, \mathbf{y} \notin \{\mathbf{y}_{i_j}\}_{j=1}^L\}$. The point $\mathbf{y}$ is said to be away from its external subspaces of dimension $r$ if $\min_{\mathbf{H} \in \mathcal{H}_{\mathbf{y},r}} d(\mathbf{y}, \mathbf{H}) > 0$. All the data points in $\mathbf{Y}^{(k)}$ are said to be away from the external subspaces if each of them is away from the its associated external spaces.

We also need the definitions related to the spectrum of $\mathbf{X}$ and $\mathbf{Y}$, which are defined as follows. In the following analysis, we employ $\boldsymbol{\beta}$ to denote the sparse code of datum $\mathbf{x}_i$ so that a simpler notation other than $\mathbf{Z}^i$ is dedicated to our analysis.

**Definition 3.3.** The minimum restricted eigenvalue of the uncorrupted data is defined as

$$\sigma_{\mathbf{Y},r} := \min_{\boldsymbol{\beta}:\|\boldsymbol{\beta}\|_0=r, \text{rank}(\mathbf{Y}_{\boldsymbol{\beta}})=\|\boldsymbol{\beta}\|_0} \sigma_{\min}(\mathbf{Y}_{\boldsymbol{\beta}})$$

for $r \ge 1$. In addition, the normalized minimum restricted eigenvalue of the uncorrupted data is defined by

$$\bar{\sigma}_{\mathbf{Y},r} := \frac{\sigma_{\mathbf{Y},r}}{\sqrt{r}}.$$

Moreover, the following quantities are defined for our analysis. We define

$$\tau_0 := \frac{2\delta\sqrt{r^*}}{\sigma_{\mathbf{X}}^*} + \tau_1, \quad (6)$$

where

$$\tau_1 := \frac{\delta}{\bar{\sigma}_{\mathbf{Y}}^* - \delta}, \quad \sigma_{\mathbf{X}}^* := \sigma_{\min}(\mathbf{X}_{\boldsymbol{\beta}^*}), \quad (7)$$

with $\delta < \bar{\sigma}_{\mathbf{Y}}^*$, and $\bar{\sigma}_{\mathbf{Y}}^*$ is defined as

$$\bar{\sigma}_{\mathbf{Y}}^* := \min_{1 \le r < r^*} \bar{\sigma}_{\mathbf{Y},r}. \quad (8)$$

Now we present our main result on noisy $\ell^0$-SSC.

**Theorem 3.1.** (Subspace detection property holds for noisy $\ell^0$-SSC) Let nonzero vector $\boldsymbol{\beta}^*$ be an optimal solution to the noisy $\ell^0$-SSC problem (5) for point $\mathbf{x}_i$ with $\|\boldsymbol{\beta}^*\|_0 = r^* \geq 1$, and $c^* := \|\mathbf{x}_i - \mathbf{X}\boldsymbol{\beta}^*\|_2$. Suppose $\mathbf{Y}$ is in general position, $\mathbf{y}_i \in \mathcal{S}_k$ for some $1 \leq k \leq K$, $\delta < \bar{\sigma}_{\mathbf{Y}}^*$, $\lambda > \tau_0$, $\mathbf{B}(\mathbf{y}_i, \delta + c^* + \frac{2\delta\sqrt{r^*}}{\sigma_{\mathbf{X}}^*}) \cap \mathbf{H} = \emptyset$ for any $\mathbf{H} \in \mathcal{H}_{\mathbf{y}_i, r^*}$. Then the subspace detection property holds for $\mathbf{x}_i$ with $\boldsymbol{\beta}^*$. Here $\tau_0, \tau_1, \bar{\sigma}_{\mathbf{Y}}^*$ and $\sigma_{\mathbf{X}}^*$ are defined in (6), (7) and (8).

**Remark 3.2.** When $\delta = 0$ and there is no noise in the data $\mathbf{X}$, the conditions for the correctness of noisy $\ell^0$-SSC in Theorem 3.1 almost reduce to that for noiseless $\ell^0$-SSC. To see this, the conditions are reduced to $\mathbf{B}(\mathbf{y}_i, c^*) \cap \mathbf{H} = \emptyset$, which are exactly the conditions required by noiseless $\ell^0$-SSC in Lemma 1.1 in the supplementary, namely data are away from the external subspaces by choosing $\lambda \to 0$ and it follows that $c^* = 0$.

While Theorem 3.1 establishes geometric conditions under which the subspace detection property holds for noisy $\ell^0$-SSC, it can be seen that these conditions are often coupled with an optimal solution $\boldsymbol{\beta}^*$ to the noisy $\ell^0$-SSC problem (5). In the following theorem, the correctness of noisy $\ell^0$-SSC is guaranteed in terms of $\lambda$, the weight for the $\ell^0$ regularization term in (5), and the geometric conditions independent of an optimal solution to (5).

Let $M_i > 0$ be the minimum distance between $\mathbf{y}_i \in \mathcal{S}_k$ and its external subspaces when $\mathbf{y}_i$ is away from its external subspaces of dimension $r$, that is,

$$M_i := \min\{d(\mathbf{y}_i, \mathbf{H}) : \mathbf{H} \in \mathcal{H}_{\mathbf{y}_i, d_k}\}, \qquad (9)$$

The following two quantities related to the spectrum of clean and noisy data, $\mu_r$ and $\sigma_{\mathbf{X}, r}$, are defined as follows with $r > 1$ for the analysis in Theorem 3.3.

$$\mu_r := \frac{\delta}{\min_{1 \leq r' < r} \bar{\sigma}_{\mathbf{Y}, r} - \delta}, \qquad (10)$$

$$\sigma_{\mathbf{X}, r} := \min\{\sigma_{\min}(\mathbf{X}_{\boldsymbol{\beta}}) : 1 \leq \|\boldsymbol{\beta}\|_0 \leq r\} \qquad (11)$$

**Theorem 3.3.** (Subspace detection property holds for noisy $\ell^0$-SSC under deterministic model with conditions in terms of $\lambda$) Let nonzero vector $\boldsymbol{\beta}^*$ be an optimal solution to the noisy $\ell^0$-SSC problem (5) for point $\mathbf{x}_i$ with $\|\boldsymbol{\beta}^*\|_0 = r^* > 1$, $n_k \geq d_k + 1$ for every $k \in [K]$, and there exists $1 < r_0 \leq \lfloor \frac{1}{\lambda} \rfloor$ such that $r^* \leq r_0$. Suppose $\mathbf{Y}$ is in general position, $\mathbf{y}_i \in \mathcal{S}_k$ for some $1 \leq k \leq K$, $\delta < \bar{\sigma}_{\mathbf{Y}}^*$, and $M_{i, \delta} := M_i - \delta$. Suppose

$$M_{i, \delta} > \frac{2\delta}{\sigma_{\mathbf{X}, r_0}}, \qquad (12)$$

and

$$\mu_{r_0} < 1 - \frac{2\delta}{\sigma_{\mathbf{X}, r_0}}. \qquad (13)$$

Then if

$$\lambda_0 < \lambda < 1, \qquad (14)$$

where $\lambda_0 := \max\{\lambda_1, \lambda_2\}$ and

$$\lambda_1 := \inf\{0 < \lambda < 1 : \sqrt{1 - \lambda} + \frac{2\delta}{\sigma_{\mathbf{X}, r_0}\sqrt{\lambda}} < M_{i, \delta}\}, \qquad (15)$$

$$\lambda_2 := \inf\{0 < \lambda < 1 : \lambda - \frac{2\delta}{\sigma_{\mathbf{X}, r_0}}\frac{1}{\sqrt{\lambda}} > \mu_{r_0}\}, \qquad (16)$$

the subspace detection property holds for $\mathbf{x}_i$ with $\boldsymbol{\beta}^*$. Here $M_i, \mu_{r_0}, \sigma_{\mathbf{X}, r_0}$ are defined in (9), (10), (11) respectively.

**Remark 3.4.** The two conditions (12) and (13) are induced by two conditions, $\mathbf{B}(\mathbf{y}_i, \delta + c^* + \frac{2\delta\sqrt{r^*}}{\sigma_{\mathbf{X}}^*}) \cap \mathbf{H} = \emptyset$ for any $\mathbf{H} \in \mathcal{H}_{\mathbf{y}_i, d_k}$ and $\lambda > \tau_0$ respectively, which are required by Theorem 3.1. Note that when (12) and (13) hold, $\lambda_1$ and $\lambda_2$ can always be chosen in accordance with (15) and (16).

**Remark 3.5.** It can be observed from condition (14) that noisy $\ell^0$-SSC encourages sparse solution by a relatively large $\lambda$ so as to guarantee the subspace detection property. This theoretical finding is consistent with the empirical study shown in the experimental results.

## 3.2 NOISY $\ell^0$-SSC: RANDOMIZED ANALYSIS

The correctness of noisy $\ell^0$-SSC is analyzed under the semi-random model that the data in subspace $\mathcal{S}^{(k)}$ are i.i.d. according to the uniform distribution on the unit sphere, $\mathbb{S}^{d_k - 1}$, of $\mathbb{R}^{d_k}$ centered at the origin for all $k \in [K]$. This setting is employed extensively in the subspace learning literature [Soltanolkotabi and Candés, 2012, Wang et al., 2013, Wang and Xu, 2013, Acharyya and Ghosh, 2015]. We then have the major theorem below stating the theoretical guarantee of the subspace detection property of noisy $\ell^0$-SSC under the semi-random model. Before stating this theorem, we introduce the following definition of subspace affinity, which is widely used in the analysis of semi-random model in the sparse subspace clustering literature.

**Definition 3.4.** (Subspace affinity) The affinity between two subspaces, $\mathcal{S}_k$ and $\mathcal{S}_l$ with $k, l \in [K]$, is defined by

$$\mathrm{aff}(\mathcal{S}_k, \mathcal{S}_l) = \sqrt{\sum_{t=1}^{\min\{k, l\}} \cos^2 \theta_{kl}^{(t)}},$$

where $\cos \theta_{kl}^{(t)}$ is the $t$-th canonical angle between $\mathcal{S}_k$ and $\mathcal{S}_l$ defined in Soltanolkotabi and Candés [2012]. Let $\mathbf{U}^{(k)}$ and $\mathbf{U}^{(l)}$ be the orthonormal basis for $\mathcal{S}_k$ and $\mathcal{S}_l$ respectively, then

$$\cos \theta_{kl}^{(t)} = \sup_{\mathbf{u} \in \mathcal{S}_k, \mathbf{v} \in \mathcal{S}_l} \frac{\mathbf{u}^\top \mathbf{v}}{\|\mathbf{u}\|_2 \|\mathbf{v}\|_2} = \frac{\mathbf{u}^{t\top} \mathbf{v}^t}{\|\mathbf{u}^t\|_2 \|\mathbf{v}^t\|_2},$$

with orthogonality: $\mathbf{u}^\top \mathbf{u}^j = 0$, $\mathbf{v}^\top \mathbf{v}^j = 0$, $j = 1, \ldots, t - 1$. It can be verified that $\mathrm{aff}(\mathcal{S}_k, \mathcal{S}_l) = \|\mathbf{U}^{(k)\top} \mathbf{U}^{(l)}\|_F$.

**Theorem 3.6.** (Subspace detection property holds for noisy $\ell^0$-SSC under semi-random model with conditions in terms of $\lambda$) Under the semi-random model, let nonzero vector $\boldsymbol{\beta}^*$ be an optimal solution to the noisy $\ell^0$-SSC problem (5) for point $\mathbf{x}_i$ with $\|\boldsymbol{\beta}^*\|_0 = r^* > 1$, $n_k \geq d_k + 1$ for every $1 \leq k \leq K$, and there exists $1 < r_0 \leq \lfloor \frac{1}{\lambda} \rfloor$ such that $r^* \leq r_0$. Suppose $c_1 > 0$ is an arbitrary small constant, $\varepsilon_0, \varepsilon_1 > 0$ be small constants, and $d_k$ is large enough such that $d_k \geq \lfloor \frac{1}{\lambda} \rfloor$, $2d_k^{-0.05} + 2d_k^{-0.1} \leq \varepsilon_0$ and $\sqrt{\frac{1}{\lambda d_k}} + \sqrt{\frac{2}{\lambda d_k} \log \frac{e n_k}{r_0}} \leq \varepsilon_1$ hold for all $k \in [K]$. Define

$$\sigma'_{\min} := \frac{1}{1 + \varepsilon_0} \left( 1 - \sqrt{c_1} - \varepsilon_1 \right), \qquad (17)$$

$c := \sqrt{\frac{\sigma'^2_{\min} - (r_0 - 1) \mathrm{aff}(\mathcal{S}_{t_1}, \mathcal{S}_{t_2})}{r_0}}$. For $t > 0$ such that $\frac{1}{d_{\max}} - 2t\sqrt{1 - \frac{1}{d_{\max}}} - t^2 > 0$, suppose

$$\max_{t_1, t_1 \in [K], t_1 \neq t_2} \mathrm{aff}(\mathcal{S}_{t_1}, \mathcal{S}_{t_2}) < \frac{\sigma'^2_{\min}}{r_0 - 1}, \qquad (18)$$

$$\delta < c, \qquad (19)$$

$$\delta + \frac{2\delta}{\sqrt{r_0}(c - \delta)} \leq \frac{1}{d_{\max}} - 2t\sqrt{1 - \frac{1}{d_{\max}}} - t^2, \quad (20)$$

$$\frac{\delta}{c - \delta} + \frac{2\delta}{\sqrt{r_0}(c - \delta)} < 1, \qquad (21)$$

$$\lambda'_0 < \lambda < 1, \qquad (22)$$

where $M := \sup_{0 \leq t < 1} \frac{2t^3 \arccos t}{\pi} < 1$, $\lambda'_0 := \max\{\lambda'_1, \lambda'_2\}$,

$$\lambda'_1 := \inf\{0 < \lambda < 1 : \sqrt{1 - \lambda} + \frac{2\delta}{\sqrt{r_0}(c - \delta)\sqrt{\lambda}}$$
$$< \frac{1}{d_{\max}} - 2t\sqrt{1 - \frac{1}{d_{\max}}} - t^2 - \delta\},$$

$$\lambda'_2 := \inf\{0 < \lambda < 1 : \lambda - \frac{2\delta}{\sqrt{r_0}(c - \delta)} \frac{1}{\sqrt{\lambda}} > \frac{\delta}{c - \delta}\}.$$

When the conditions in Lemma 1.5 of the supplementary hold for all $k \in [K]$ and every point $\mathbf{y} \in \mathbf{Y}^{(k)}$, then with probability at least $1 - \sum_{k=1}^{K} \left( \exp(-c_1 d_k) + 2n_k \exp\left(-d_k^{0.9}\right) \right) - 8 \sum_{k=1}^{K} n_k \exp(-\frac{d_k t^2}{2})$, the subspace detection property holds for $\mathbf{x}_i$ with $\boldsymbol{\beta}^*$ for all $i \in [n]$.

**Remark 3.7** (Advantage of Noisy $\ell^0$-SSC in terms of Subspace Affinity). It is well known that the difficulty of achieving the subspace detection property increases with larger affinity between subspaces, that is, the subspace are closer to each other. Our analysis reveals the significant advantage of noisy $\ell^0$-SSC over $\ell^1$-SSC in terms of the maximum subspace affinity. To the best of our knowledge, the best theoretical result of $\ell^1$-SSC, including its geometrical analysis [Soltanolkotabi and Candés, 2012] and the subsequent works on noisy or dimensionality-reduced data [Wang and

Xu, 2013, Wang et al., 2015], requires that the maximum subspace affinity satisfies

$$\max_{t_1, t_1 \in [K], t_1 \neq t_2} \mathrm{aff}(\mathcal{S}_{t_1}, \mathcal{S}_{t_2}) < \sqrt{d_0} \cdot \frac{\bar{c}\sqrt{\log \rho}}{8\sqrt{2} \log n}, \qquad (23)$$

under the setting in Soltanolkotabi and Candés [2012] that $d_k = d_0$ and $n_k = \rho d_0 + 1$ for all $k \in [K]$, so that $n = K(\rho d_0 + 1)$. When $n > \exp(d_0^\tau)$ for $\tau \in (0.5, 0.9)$, then (23) requires $\max_{t_1, t_1 \in [K], t_1 \neq t_2} \mathrm{aff}(\mathcal{S}_{t_1}, \mathcal{S}_{t_2}) \to 0$ when $d_0 \to \infty$, while the condition of noisy $\ell^0$-SSC, (18), only requires that $\max_{t_1, t_1 \in [K], t_1 \neq t_2} \mathrm{aff}(\mathcal{S}_{t_1}, \mathcal{S}_{t_2}) < \frac{\sigma'^2_{\min}}{r_0 - 1}$ when $d_0$ is sufficiently large. Such less restive condition on the maximum subspace affinity reveals the theoretical advantage of noisy $\ell^0$-SSC over $\ell^1$-SSC.

# 4 NOISY $\ell^0$-SSC ON DIMENSIONALITY REDUCED DATA: NOISY-DR-$\ell^0$-SSC

Albeit the theoretical guarantee and compelling empirical performance of noisy $\ell^0$-SSC to be shown in the experimental results, the computational cost of noisy $\ell^0$-SSC is high with the high dimensionality of the data. In this section, we propose Noisy Dimensionality Reduced $\ell^0$-SSC (Noisy-DR-$\ell^0$-SSC) which performs noisy $\ell^0$-SSC on dimensionality reduced data. The theoretical guarantee on the correctness of Noisy-DR-$\ell^0$-SSC under the deterministic model as well as its empirical performance are presented.

## 4.1 METHOD

Noisy-DR-$\ell^0$-SSC performs subspace clustering by the following two steps: 1) obtain the dimension reduced data $\tilde{\mathbf{X}} = \mathbf{P}\mathbf{X}$ with a linear transformation $\mathbf{P} \in \mathbb{R}^{p \times d}$ ($p < d$). 2) perform noisy $\ell^0$-SSC on the compressed data $\tilde{\mathbf{X}}$:

$$\min_{\tilde{\boldsymbol{\beta}} \in \mathbb{R}^n, \tilde{\boldsymbol{\beta}}_i = 0} L(\tilde{\boldsymbol{\beta}}) = \|\tilde{\mathbf{x}}_i - \tilde{\mathbf{X}}\boldsymbol{\beta}\|_2^2 + \tilde{\lambda}\|\tilde{\boldsymbol{\beta}}\|_0. \qquad (24)$$

If $p < d$, Noisy-DR-$\ell^0$-SSC operates on the compressed data $\tilde{\mathbf{X}}$ rather than on the original data, so that the efficiency is improved. We introduce two types of random projection for Noisy-DR-$\ell^0$-SSC in the following two subsections.

## 4.2 RANDOMIZED LOW-RANK APPROXIMATION

High-dimensional data often exhibits low-dimensional structures, which often leads to low-rankness of the data matrix. Intuitively, if the data is low rank, then it could be safe to perform noisy $\ell^0$-SSC on its dimensionality reduced version by the linear projection $\mathbf{P}$, and it is expected that $\mathbf{P}$ can preserve the information of the subspaces contained in the original data as much as possible, while effectively removing uninformative dimensions. To this end, we propose to

choose $\mathbf{P}$ as a random projection induced by randomized low-rank approximation of the data.

The merit of random projection (RP) is highlighted by the celebrated Johnson-Lindenstrauss Lemma [Johnson and Lindenstrauss, 1984]. In the past 20 years or more, RP has been used extensively in dimension reduction, approximate near neighbor search, compressed sensing, computational biology, etc [Dasgupta, 2000, Bingham and Mannila, 2001, Buhler, 2001, Achlioptas, 2003, Fern and Brodley, 2003, Datar et al., 2004, Candès et al., 2006, Donoho, 2006, Freund et al., 2007, Li, 2007, 2017, 2019]. In particular, RP has been employed to accelerate numerical matrix computation and matrix optimization problems, including matrix decomposition [Frieze et al., 2004, Drineas et al., 2004, Sarlós, 2006, Drineas et al., 2006, 2008, Mahoney and Drineas, 2009, Drineas et al., 2011, Lu et al., 2013].

Formally, a random matrix $\mathbf{T} \in \mathbb{R}^{n \times p}$ is generated such that each element $\mathbf{T}_{ij}$ is sampled independently according to the Gaussian distribution $\mathcal{N}(0, 1)$. QR decomposition is then performed on $\mathbf{XT}$ to obtain the basis of its column space, namely $\mathbf{XT} = \mathbf{QR}$ where $\mathbf{Q} \in \mathbb{R}^{d \times p}$ is an orthogonal matrix of rank $p$ and $\mathbf{R} \in \mathbb{R}^{p \times p}$ is an upper triangle matrix. The columns of $\mathbf{Q}$ form the orthogonal basis for the sample matrix $\mathbf{XT}$. An approximation of $\mathbf{X}$ is then obtained by projecting $\mathbf{X}$ onto the column space of $\mathbf{XT}$: $\mathbf{QQ}^\top \mathbf{X} = \mathbf{QW} = \widehat{\mathbf{X}}$ where $\mathbf{W} = \mathbf{Q}^\top \mathbf{X} \in \mathbb{R}^{p \times n}$. In this manner, a randomized low-rank decomposition of $\mathbf{X}$ is achieved by

$$\widehat{\mathbf{X}} = \mathbf{QW}.$$

It is proved that the low rank approximation $\bar{\mathbf{X}}$ is close to $\mathbf{X}$ in spectral norm [Halko et al., 2011]. We present probabilistic result in Theorem 4.1 on the correctness of Noisy-DR-$\ell^0$-SSC using the random projection induced by randomized low-rank decomposition of the data $\mathbf{X}$, namely $\mathbf{P} = \mathbf{Q}^\top$. In the sequel, $\tilde{\mathbf{x}} = \mathbf{Px}$ for any $\mathbf{x} \in \mathbb{R}^n$. To guarantee the subspace detection property on the dimensionality-reduced data $\tilde{\mathbf{X}}$, it is crucial to ensure that the conditions, such as (12) (13) in Theorem 3.3, still hold after linear transformation.

Each subspace $\mathcal{S}_k$ is transformed into $\tilde{\mathcal{S}}_k = \mathbf{P}(\mathcal{S}_k)$ with dimension $\tilde{d}_k$. We denote by $\tilde{\boldsymbol{\beta}}^*$ an optimal solution to (24), and define $C_{p,p_0} := \left(1 + 17\sqrt{1 + \frac{p_0}{p - p_0}}\right)\sigma_{p_0 + 1} + \frac{8\sqrt{p}}{p - p_0 + 1}\left(\sum_{j > p_0} \sigma_j^2\right)^{\frac{1}{2}}$ with $p_0 \geq 2$. We also define the following quantities for the convenience of our analysis, which correspond to $M_i$, $\bar{\sigma}_{\mathbf{Y},r}$, $\sigma_{\mathbf{X},r}$ and $\mu_r$ used in the analysis on the original data:

$$\tilde{M}_i := \min\{d(\tilde{\mathbf{y}}_i, \mathbf{H}): \mathbf{H} \in \mathcal{H}_{\tilde{\mathbf{y}}_i, \tilde{d}_k}\}, \qquad (25)$$

where $\mathcal{H}_{\tilde{\mathbf{y}}_i, \tilde{d}_k}$ is all the external subspaces of $\tilde{\mathbf{y}}_i$ with dimension no greater than $\tilde{d}_k$ in the transformed space by $\mathbf{P}$,

$$\bar{\sigma}_{\tilde{\mathbf{Y}},r} := \min_{\boldsymbol{\beta}: \|\boldsymbol{\beta}\|_0 = r, \text{rank}(\tilde{\mathbf{Y}}_{\boldsymbol{\beta}}) = \|\boldsymbol{\beta}\|_0} \sigma_{\min}(\tilde{\mathbf{Y}}_{\boldsymbol{\beta}}), \qquad (26)$$

$$\sigma_{\tilde{\mathbf{X}},r} := \min\{\sigma_{\min}(\tilde{\mathbf{X}}_{\boldsymbol{\beta}}): 1 \leq \|\boldsymbol{\beta}\|_0 \leq r\}, \qquad (27)$$

$$\tilde{\mu}_r := \frac{\delta}{\min_{1 \leq r' < r} \bar{\sigma}_{\tilde{\mathbf{Y}},r} - \delta}. \qquad (28)$$

**Theorem 4.1.** (Subspace detection property holds for Noisy-DR-$\ell^0$-SSC under deterministic model) Let nonzero vector $\boldsymbol{\beta}^*$ be an optimal solution to the noisy $\ell^0$-SSC problem (5) for point $\mathbf{x}_i$ with $\|\boldsymbol{\beta}^*\|_0 = r^* > 1$, $n_k \geq d_k + 1$ for every $1 \leq k \leq K$, and there exists $1 < r_0 \leq d_k$ such that $r^* \leq r_0 \leq \lfloor \frac{1}{\lambda} \rfloor$. Suppose $\mathbf{Y}$ is in general position, $\delta < \min_{1 \leq r < r_0} \bar{\sigma}_{\mathbf{Y},r}$, and $\tilde{M}_{i,\delta} := \tilde{M}_i - \delta$. Furthermore, suppose the following conditions hold:

(i) $C_{p,p_0} + 2\delta\sqrt{\tilde{d}_{\max}} < \min_{k=1,\ldots,K} \sigma_{\mathbf{Y}}^{(k)}$, where $\tilde{d}_{\max} := \max_k \tilde{d}_k$, $\sigma_{\mathbf{Y}}^{(k)} := \min\{\sigma_{\min}(\mathbf{A}): \mathbf{A} \subseteq \mathbf{Y}^{(k)}, \mathbf{A} \in \mathbb{R}^{d \times n'}, n' \leq \tilde{d}_k\}$,

(ii) $\delta(1 + 2\sqrt{r_0}) < \min_{1 \leq r < r_0} \bar{\sigma}_{\mathbf{Y},r} - C_{p,p_0}$,

(iii) $\min_{1 \leq r \leq \tilde{d}_k} \sigma_{\mathbf{Y},r} > C_{p,p_0} - 2\delta\sqrt{\tilde{d}_k}$ and

$$M_i - C_{p,p_0}(1 + \frac{1}{\min_{1 \leq r \leq \tilde{d}_k} \sigma_{\mathbf{Y},r} - C_{p,p_0} - 2\delta\sqrt{\tilde{d}_k}})$$
$$> \delta + \frac{2\delta}{\sigma_{\mathbf{X},r_0} - C_{p,p_0}},$$

for all $\mathbf{y}_i \in \mathcal{S}_k$ and $1 \leq k \leq K$,

(iv) $\min_{1 \leq r < r_0} \bar{\sigma}_{\mathbf{Y},r_0} > C_{p,p_0} - 2\delta\sqrt{r_0} - \delta$ and

$$\frac{\delta}{\min_{1 \leq r < r_0} \bar{\sigma}_{\mathbf{Y},r_0} - C_{p,p_0} - 2\delta\sqrt{r_0} - \delta}$$
$$< 1 - \frac{2\delta}{\sigma_{\mathbf{X},r_0} - C_{p,p_0}}.$$

If $\tilde{\lambda}_0 < \tilde{\lambda} < 1$, where $\tilde{\lambda}_0 = \max\{\tilde{\lambda}_1, \tilde{\lambda}_2\}$ and

$$\tilde{\lambda}_1 = \inf\{0 < \tilde{\lambda} < 1: \sqrt{1 - \tilde{\lambda}} + \frac{2\delta}{\sigma_{\tilde{\mathbf{X}},r_0}\sqrt{\tilde{\lambda}}} < \tilde{M}_{i,\delta}\}, \qquad (29)$$

$$\tilde{\lambda}_2 = \inf\{0 < \tilde{\lambda} < 1: \tilde{\lambda} - \frac{2\delta}{\sigma_{\mathbf{X},r_0}} \frac{1}{\sqrt{\tilde{\lambda}}} > \tilde{\mu}_{r_0}\}, \qquad (30)$$

then with probability at least $1 - 6e^{-p}$, the subspace detection property holds for $\tilde{\mathbf{x}}_i$ with $\tilde{\boldsymbol{\beta}}^*$. Here $\tilde{M}_i$, $\tilde{\mu}_r$ and $\tilde{\sigma}_{\tilde{\mathbf{X}},r_0}$ are defined in (25), (28) and (27) respectively.

## 4.3 VERY SPARSE RANDOM PROJECTIONS

In this subsection, we study the case when the linear transformation $\mathbf{P}$ for the dimensionality reduced $\ell^0$-SSC problem (24) is a sparse matrix [Charikar et al., 2004, Cormode and Muthukrishnan, 2005, Li, 2007, Weinberger et al., 2009, Gilbert and Indyk, 2010, Li et al., 2011, Nelson and Nguyen, 2013, Li and Zhao, 2022]. In particular, we choose $\mathbf{P}$ such

that each column of $\mathbf{P}$ only has 1 nonzero element, in a fashion known as "count-sketch" [Charikar et al., 2004]. Weinberger et al. [2009] applied count-sketch as a dimension reduction tool for machine learning. The work of [Li et al., 2011], in addition to developing hash learning algorithm based on minwise hashing, also provided the thorough theoretical analysis for count-sketch in the context of estimating inner products. The conclusion from Li et al. [2011] is that, to estimate inner products, we should use count-sketch (or very sparse random projections [Li, 2007]) instead of the original (dense) random projections, because count-sketch is not only computationally much more efficient but also (slightly) more accurate, as far as the task of similarity estimation is concerned.

Using those nice theoretical properties of count-sketch projections, we have the following theorem about the correctness of Noisy-DR-$\ell^0$-SSC when $\mathbf{P}$ has only 1 nonzero element in each column. For breveity, we name such a projection matrix to be "CSP".

**Theorem 4.2.** (Subspace detection property holds for Noisy-DR-$\ell^0$-SSC under deterministic model with $\mathbf{P}$ being the CSP) Let nonzero vector $\boldsymbol{\beta}^*$ be the optimal solution to the noisy $\ell^0$-SSC problem (5) for point $\mathbf{x}_i$ with $\|\boldsymbol{\beta}^*\|_0 = r^*$, $n_k \geq d_k + 1$ for every $1 \leq k \leq K$, and there exists $1 < r_0 \leq \lfloor \frac{1}{\lambda} \rfloor$ such that $1 < r^* \leq r_0$. Suppose $\mathbf{Y}$ is in general position, $\mathbf{y}_i \in \mathcal{S}_k$ for some $1 \leq k \leq K$, $\delta < \min_{1 \leq r < r_0} \bar{\sigma}_{\mathbf{Y},r}$. Let $M_{i,\delta} := M_i - \delta$, $\varepsilon$ be a positive number such that $0 < \varepsilon \leq 1$. Suppose

$$M_{i,\delta} > \frac{2(1+\varepsilon)^3 \delta}{\sigma_{\mathbf{X},r_0}}, \tag{31}$$

$$\mu_{r,\varepsilon} := \frac{\delta}{\frac{\min_{1 \leq r' < r} \bar{\sigma}_{\mathbf{Y},r}}{(1+\varepsilon)^2} - \delta} < 1 - \frac{2(1+\varepsilon)^2 \delta}{\sigma_{\mathbf{X},r_0}}. \tag{32}$$

Then if $\tilde{\lambda}_0 < \tilde{\lambda} < 1$, where $\tilde{\lambda}_0 := \max\{\lambda_1, \lambda_2\}$ and

$$\lambda_1 := \inf\{0 < \tilde{\lambda} < 1 : \sqrt{1 + \varepsilon - \tilde{\lambda}} + \frac{2\delta}{\sigma_{\tilde{\mathbf{X}},r_0}\sqrt{\tilde{\lambda}}} < M_{i,\delta}\}, \tag{33}$$

$$\lambda_2 := \inf\{0 < \lambda < 1 : \lambda - \frac{2\delta}{\sigma_{\tilde{\mathbf{X}},r_0}} \frac{1}{\sqrt{\lambda}} > \tilde{\mu}_{r_0}\}, \tag{34}$$

then with probability at least $1 - K\delta'$ for all $\delta' \in (0, \frac{1}{K})$, the subspace detection property holds for $\tilde{\mathbf{x}}_i$ with $\tilde{\boldsymbol{\beta}}^*$. Here $\tilde{\mu}_{r_0}$ and $\sigma_{\tilde{\mathbf{X}},r_0}$ are defined in (28) and (27) respectively. $\tilde{\boldsymbol{\beta}}^*$ is the optimal solution to (24) with $\mathbf{P}$ being the CSP described in the beginning of this subsection with $p \geq \frac{d_{\max}^2 + d_{\max}}{\delta'(2\varepsilon - \varepsilon^2)^2}$.

### 4.4 THE ALGORITHM OF NOISY-DR-$\ell^0$-SSC

We denote by Noisy-DR-$\ell^0$-SSC-LR the Noisy-DR-$\ell^0$-SSC with random projection induced by randomized low-rank

approximation in Section 4.2, and denote by Noisy-DR-$\ell^0$-SSC-CSP the Noisy-DR-$\ell^0$-SSC with CSP serving as the random projection in Section 4.3.

---

**Algorithm 1** Noisy Dimensionality Reduced $\ell^0$-Sparse Subspace Clustering by Randomized Low-Rank Approximation (Noisy-DR-$\ell^0$-SSC-LR)

1: Generate a Gaussian random matrix $\mathbf{T} \in \mathbb{R}^{n \times p}$ where each element $\mathbf{T}_{ij}$ is sampled independently according to the standard Gaussian distribution $\mathcal{N}(0, 1)$
2: Perform QR decomposition on $\mathbf{XT}$, $\mathbf{XT} = \mathbf{QR}$ where $\mathbf{Q} \in \mathbb{R}^{d \times p}$
3: Set the linear transformation $\mathbf{P} = \mathbf{Q}^\top$, and obtain the dimensionality reduced data $\tilde{\mathbf{X}} = \mathbf{PX}$
4: Perform noisy $\ell^0$-SSC on $\tilde{\mathbf{X}}$ using Algorithm 1

---

Noisy-DR-$\ell^0$-SSC-LR is described by Algorithm 1. The algorithm of Noisy-DR-$\ell^0$-SSC-CSP is similar to Algorithm 1 except that CSP serves as the random projection $\mathbf{P}$. Algorithm 1 in Section 4 of the supplementary describes how to solve the noisy $\ell^0$-SSC problem (5).

## 5 EXPERIMENTS

We demonstrate the performance of Noisy-DR-$\ell^0$-SSC-LR and Noisy-DR-$\ell^0$-SSC-CSP, with comparison to other competing clustering methods including K-means (KM), Spectral Clustering (SC), noisy SSC, Sparse Manifold Clustering and Embedding (SMCE) [Elhamifar and Vidal, 2011] and SSC-OMP [Dyer et al., 2013] in this section. We will use Noisy-DR-$\ell^0$-SSC to refer to its two variants. With the coefficient matrix $\mathbf{Z}$ obtained by the optimization of noisy $\ell^0$-SSC or Noisy-DR-$\ell^0$-SSC, a sparse similarity matrix is built by $\mathbf{W} = \frac{|\mathbf{Z}| + |\mathbf{Z}^\top|}{2}$, and spectral clustering is performed on $\mathbf{W}$ to obtain the clustering results. Two measures are used to evaluate the performance of different clustering methods, i.e. the Accuracy (AC) and the Normalized Mutual Information (NMI) [Zheng et al., 2004].

We use randomized rank-$p$ decomposition of the data matrix in Noisy-DR-$\ell^0$-SSC-LR with $p = \frac{\min\{d,n\}}{10}$. It can be observed that noisy $\ell^0$-SSC and Noisy-DR-$\ell^0$-SSC always achieve better performance than other methods in Table 2, including the noisy SSC on dimensionality reduced data (Noisy DR-SSC) [Wang et al., 2015]. Note that noisy $\ell^0$-SSC has the same performance as $\ell^0$-SSC [Yang et al., 2018]. Throughout all the experiments we find that the best clustering accuracy is achieved whenever $\lambda$ is chosen by $0.5 < \lambda < 0.95$, justifying our theoretical finding claimed in Remark 3.5 and (14) in Theorem 3.3. For all the methods that involve random projection, we conduct the experiments for 30 times and report the average performance. Note that the cluster accuracy of SSC-OMP on the extended Yale-B data set is reported according to You et al. [2016]. We randomly sample 1000 images from each class of the MNIST data set so as to collect a total number of 10000 images on

Table 2: Clustering results on various data sets, with the best three results in bold.

| Data Set | Measure | KM | SC | Noisy SSC | Noisy DR-SSC | SMCE | SSC-OMP | Noisy $\ell^0$-SSC | Noisy-DR-$\ell^0$-SSC-LR | Noisy-DR-$\ell^0$-SSC-CSP |
|---|---|---|---|---|---|---|---|---|---|---|
| COIL-20 | AC | 0.6554 | 0.4278 | 0.7854 | 0.7764 | 0.7549 | 0.3389 | **0.8472**± 0.0031 | **0.8479**± 0.0023 | **0.8472**± 0.0019 |
| | NMI | 0.7630 | 0.6217 | 0.9148 | 0.9219 | 0.8754 | 0.4853 | **0.9428**± 0.0082 | **0.9433**± 0.0063 | **0.9429**± 0.0037 |
| COIL-100 | AC | 0.4996 | 0.2835 | 0.5275 | 0.5013 | 0.5639 | 0.1667 | **0.7683**± 0.0020 | **0.7039**± 0.0087 | **0.7046**± 0.0083 |
| | NMI | 0.7539 | 0.5923 | 0.8041 | 0.8019 | 0.8064 | 0.3757 | **0.9182**± 0.0096 | **0.8706**± 0.0109 | **0.8708**± 0.0117 |
| Yale-B | AC | 0.0954 | 0.1077 | 0.7850 | 0.7255 | 0.3293 | 0.7789 | **0.8480**± 0.0091 | **0.8231**± 0.0173 | **0.8318**± 0.0112 |
| | NMI | 0.1258 | 0.1485 | 0.7760 | 0.7311 | 0.3812 | 0.7024 | **0.8612**± 0.0072 | **0.8533**± 0.0294 | **0.8593**± 0.0133 |
| MPIE S1 | AC | 0.1164 | 0.1285 | 0.5892 | 0.3588 | 0.1721 | 0.1695 | **0.6741**±0.0413 | **0.6741**± 0.0938 | **0.6744**± 0.0662 |
| | NMI | 0.5049 | 0.5292 | 0.7653 | 0.6806 | 0.5514 | 0.3395 | **0.8622**± 0.0533 | **0.8622**± 0.0834 | **0.8548**± 0.0931 |
| MPIE S2 | AC | 0.1315 | 0.1410 | 0.6994 | 0.4611 | 0.1898 | 0.2093 | **0.7527**± 0.0115 | **0.7533**± 0.0596 | **0.7517**± 0.0813 |
| | NMI | 0.4834 | 0.5128 | 0.8149 | 0.7086 | 0.5293 | 0.4292 | **0.8939**± 0.0389 | **0.8926**± 0.0742 | **0.8910** ± 0.0454 |
| MPIE S3 | AC | 0.1291 | 0.1459 | 0.6316 | 0.4841 | 0.1856 | 0.1787 | **0.7050**± 0.0277 | **0.7123**± 0.0812 | **0.7184**± 0.1045 |
| | NMI | 0.4811 | 0.5185 | 0.7858 | 0.7340 | 0.5155 | 0.3415 | **0.8750**± 0.0157 | **0.8455**± 0.0693 | **0.8457**± 0.0913 |
| MPIE S4 | AC | 0.1308 | 0.1463 | 0.6803 | 0.5511 | 0.1823 | 0.1680 | **0.7246**± 0.0147 | **0.7137**± 0.0605 | **0.7250**± 0.0443 |
| | NMI | 0.4866 | 0.5280 | 0.8063 | 0.7955 | 0.5294 | 0.3345 | **0.8837**± 0.0212 | **0.8847**± 0.0781 | **0.8834**± 0.0517 |
| MNIST | AC | 0.5236 | 0.3504 | 0.5714 | 0.5123 | **0.6542** | 0.5561 | 0.6259 ± 0.0249 | **0.6296** ± 0.1522 | **0.6310** ± 0.1031 |
| | NMI | 0.4770 | 0.3607 | 0.6091 | 0.5026 | **0.6796** | 0.5986 | **0.6501** ± 0.0196 | 0.6440 ± 0.0259 | **0.6497** ± 0.0313 |

which clustering is performed, and the average performance of 10 random sampling is reported for this data set. The time complexity of noisy $\ell^0$-SSC and the two variants of Noisy-DR-$\ell^0$-SSC are analyzed in Section 3 of the supplementary. The actual running time of both algorithms confirms such time complexity, and we observe that Noisy-DR-$\ell^0$-SSC-LR is always 8.7 times faster than noisy $\ell^0$-SSC with the same number of iterations, and the acceleration is boosted to 9.6 times by Noisy-DR-$\ell^0$-SSC-CSP due to sparse random projections.

We further demonstrate the practical implication of our theoretical analysis for noisy $\ell^0$-SSC. As mentioned in Remark 3.5, a relatively large $\lambda$ tends to preserve the subspace detection property. This theoretical finding is consistent with the empirical study shown in this subsection. We add Gaussian noise of zero mean and different choices of variance $\sigma^2$ to the extended Yale-B data set. In Section 5 of the supplementary, Figure 2a to Figure 2f illustrate SDP violation with respect to $\lambda$ for different noise levels with $\sigma^2$ ranging over $10, 20, 30, 40, 50, 60$. The SDP violation is defined in Wang and Xu [2013] which is the percentage of pairs of data points which are mistakenly put in the same subspace by the similarity matrix $\mathbf{W}$, namely the percentage of pairs $(\mathbf{x}_i, \mathbf{x}_j)$ with nonzero $\mathbf{W}_{ij}$ while they are in fact not in the same subspace. We observe that increasing $\lambda$ effectively reduces SDP violation for noisy $\ell^0$-SSC, Noisy-DR-$\ell^0$-SSC-LR and Noisy-DR-$\ell^0$-CSP, confirming our theoretical prediction.

## 6 CONCLUSION

In this paper, we prove that noisy $\ell^0$-SSC recovers subspaces from noisy data through $\ell^0$-induced sparsity. Our results for the first time reveal the theoretical advantage of noisy $\ell^0$-SSC over its $\ell^1$ counterpart and other competing subspace clustering methods in terms of much less restrictive condition on the subspace affinity, when the size of data grows exponentially in the subspace dimension. We then propose Noisy-DR-$\ell^0$-SSC to improve the efficiency of noisy $\ell^0$-SSC, which performs noisy $\ell^0$-SSC on dimensionality reduced data and still provably recovers the underlying

subspaces. Experiments evidence the findings of our theoretical results in the robustness of noisy $\ell^0$-SSC against noise as well as the effectiveness of Noisy-DR-$\ell^0$-SSC.

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
