# OpenReview forum: "Noisy $\ell^{0}$-Sparse Subspace Clustering on Dimensionality Reduced Data"
_auai.org/UAI/2022/Conference — UAI 2022 Poster_

### Official Review · Reviewer_b3kq · 2022-04-05

**Q2(1) Originality/Novelty:** 3
**Q2(2) Significance/Impact:** 2
**Q2(3) Correctness/Technical Quality:** 3
**Q2(6) Clarity Of Writing:** 3
**Q6 Overall Score:** 7
**Q8 Confidence In Your Score:** 4

**Q1 Summary And Contributions:**

The paper proposes a theoretical analysis of L0-sparse subspace clustering (SSC) in the presence of noise. More precisely, it provides conditions under which L0-SSC is subspace preserving, that is, the matrix Z used in the self-representation X~XZ satisfies Z(i,j) \neq 0 only if the ith and jth data points belong to the same subspace.  They also provide a way to reduce the dimensionality of the data points while preserving the robustness to noise property.

**Q2 Assessment Of The Paper:**

More detailed information regarding each of these aspects is given below:

**Q2(4) Quality Of Experiments (Optional):**

3: Good: The experimental evaluation is adequate, and the results convincingly support the main claims.

**Q2(5) Reproducibility:**

2: Fair: Key resources (e.g., proofs, code, data) are unavailable but key details (e.g., proof sketches, experimental setup) are sufficiently well-described for an expert to confidently reproduce the main results.

**Q3 Main Strengths:**

The main strength is the theoretical analysis of L0-SSC. More precisely, authors analyze the following model:
min_Z ||X - XZ||_F^2 + lambda*||Z||_0 such that diag(Z) = 0,   (1)
where the columns of X are the data points. They prove that optimal solutions of (1) are subspace preserving, under appropriate conditions.

**Q4 Main Weakness:**

1) The part about the use of dimensionality reduction (DR) to reduce the computational load is less convincing/interesting to me, for two reasons:
- The use of DR is well-known in such contexts, and authors essentially use existing approaches and rely on existing analysis to compelemnt their theory of noisy L0-SSC.
- The main bottleneck of (L0-)SSC is not the dimension of the data (=d) but rather the number of data points (=n), the latter is not impacted by the DR.

2) The authors completely forget to discuss the oversegmentation issue in SSC. Taking lambda very large will make Z very sparse and therefore more prone to being subspace preserving. However, in practice, this will lead to oversegmentation.

3) The numerical experiments should be better described in the text. How was lambda chosen exactly? Did authorss also tune the lambda parameter for the competing approaches (like SSC)? Also, L0-SSC relies on PGD: how did authors initialize it? How sensitive is it to noise?




**Q5 Detailed Comments To The Authors:**

Other comments:
- Authors should better introduce the model behind SSC. A reader not familiar with SSC will not understand the model. This should be briefly recalled and motivated.
- In Definition 2.1, what is the value of lambda? How can you be sure the optimal solution of (4) is unique?
- In the theorems, authors focus on a single column of X, xi, which is fine. But then, does the lambda parameter depend on the data point xi? Sohuldit be different for different xi's? This could be clarified. Also, how can we be sure a beta^* with cardinality > 1 exists for some lambda: this seems to be assumed in the theorem, but is this always possible?


Typos/unclear parts:
- Unfinished sentence: "In the following section, we provide theoretical analysis of"
- sentences should not start with a symbol,
- Notation should be singular
- The notation y_{i_j} is not introduced and rather confusing.
- equations are part of the text and should be punctuated accordingly (e.g., 7-12-13)
- Notation B(.,.) not defined
- End of Theorem 3.6: why for all i? Don't you analyze a single i in this theorem?
- Unclear sentence: "Throughout this paper, we refer to ℓ0-SSC for noisy data with the unconstrained ℓ0-regularized problem as noisy ℓ0-SSC."

**********************************
After the reply of the authors
**********************************
The authors have properly addressed my comments, as well as most comments of other reviewers, I believe. This is overall a nice complement to the literature, in my opinion.


**Q7 Justification For Your Score:**

The theoretical analysis of L0-SSC is a nice contribution to the literature.

**Q9 Complying With Reviewing Instructions:**

1: Yes.

---

### Official Review · Reviewer_aEjY · 2022-04-13

**Q2(1) Originality/Novelty:** 3
**Q2(2) Significance/Impact:** 3
**Q2(3) Correctness/Technical Quality:** 3
**Q2(6) Clarity Of Writing:** 3
**Q6 Overall Score:** 5
**Q8 Confidence In Your Score:** 5

**Q1 Summary And Contributions:**

This paper provides a theoretical guarantee on the correctness of noisy $\ell^{0}$-SSC in terms of SDP on noisy data for the first time and proposes a Noisy-DR-$\ell^{0}$-SSC model which provably recovers the subspaces on dimensionality reduced data to improve the efficiency of noisy $\ell^{0}$-SSC.

**Q2 Assessment Of The Paper:**

More detailed information regarding each of these aspects is given below:

**Q2(4) Quality Of Experiments (Optional):**

2: Fair: The experimental evaluation is weak: important baselines are missing, or the results do not adequately support the main claims.

**Q2(5) Reproducibility:**

2: Fair: Key resources (e.g., proofs, code, data) are unavailable but key details (e.g., proof sketches, experimental setup) are sufficiently well-described for an expert to confidently reproduce the main results.

**Q3 Main Strengths:**

This paper proves that noisy $\ell^{0}$-SSC recovers subspaces from noisy data through $\ell^{0}$-induced sparsity for the first time. The theoretical proofs of these theorems are sufficiently well-described.

**Q4 Main Weakness:**

[1] The experiments are not sufficient and well described.

[2] Some motivations are not well explained.

**Q5 Detailed Comments To The Authors:**

[1] Some motivations are not well explained. For example, why does the theoretical analysis consider both the deterministic and semi-random models simultaneously? why is the dimension reduction strategy adopted?

[2] In the experiment, the competing methods are not sufficient. The original SSC method should be compared. Also, some noisy variants of LRR based subspace clustering methods should also be compared.

[3] Table 1 shows the performance of the proposed Noisy-DR- $\ell^{0}$-SSC-LR and Noisy-DR- $\ell^{0}$-SSC-OSNAP methods are worse than the Noisy-$\ell^{0}$-SSC. Does the dimension reduction make sense?

**Q7 Justification For Your Score:**

Novelty is the most important factor in my rating.

**Q9 Complying With Reviewing Instructions:**

1: Yes.

---

### Official Review · Reviewer_9ZZx · 2022-04-29

**Q2(1) Originality/Novelty:** 3
**Q2(2) Significance/Impact:** 3
**Q2(3) Correctness/Technical Quality:** 3
**Q2(6) Clarity Of Writing:** 2
**Q6 Overall Score:** 6
**Q8 Confidence In Your Score:** 3

**Q1 Summary And Contributions:**

In this paper, the authors  propose  sparse clustering methods with l_0 norm and dimensional reduction.  The proposed methods are reasonable

**Q2 Assessment Of The Paper:**

More detailed information regarding each of these aspects is given below:

**Q2(4) Quality Of Experiments (Optional):**

3: Good: The experimental evaluation is adequate, and the results convincingly support the main claims.

**Q2(5) Reproducibility:**

3: Good: Key resources (e.g., proofs, code, data) are available and key details (e.g., proofs, experimental setup) are sufficiently well-described for competent researchers to confidently reproduce the main results.

**Q3 Main Strengths:**

1.  The proposed sparse subspace clustering methods with sparsity are based on by l_0-norm compared to the original methods which are based on l_1.  The authors consider the advantages of l_0 norm in sparsity compared to l_1 norm.

2.  The authors suggested to integrate the dimensional reduction step in the procedure to dealt with high dimensional data. It is reasonable, and the selected dimensional reduction methods, random projection, would not produce  much biases. Hence I believe the results are reasonable and stable.

**Q4 Main Weakness:**

Introduction of the original methods about subspace clustering methods with sparsity is not much clear.

**Q5 Detailed Comments To The Authors:**

Though I believe the proposed methods are reasonable,  from the paper,  I cannot grasp the basic original idea of the subspace clustering methods quickly.  The paper does not provide clear introduction,  and I have to check the references firstly.

The algorithm is based on l_0 norm,  but the paper does not provide a detailed algorithm to solve optimal problems related to l_0 norm. Normally,  the computational burden for such kind optimal problem is some heavier.






**Q7 Justification For Your Score:**

The  motivations of this paper is clear and reasonable.  Compared to l_1 norm, l_0 norm has its advantages in sparsity. To high dimensional data,  making dimensional reduction firstly is necessary. The ideas of the authors are not bad.

**Q9 Complying With Reviewing Instructions:**

1: Yes.

---

### Official Review · Reviewer_jkW9 · 2022-04-30

**Q2(1) Originality/Novelty:** 2
**Q2(2) Significance/Impact:** 2
**Q2(3) Correctness/Technical Quality:** 2
**Q2(6) Clarity Of Writing:** 2
**Q6 Overall Score:** 3
**Q8 Confidence In Your Score:** 3

**Q1 Summary And Contributions:**

The paper aims to study the theoretical guarantee on the correctness of a sparse subspace method with noisy data.  It further proposes a sparse subspace clustering method based on the analysis.  The proposed method was evaluated on some standard benchmark data sets and showed superior clustering performance over some typical subspace clustering methods.


**Q2 Assessment Of The Paper:**

More detailed information regarding each of these aspects is given below:

**Q2(4) Quality Of Experiments (Optional):**

1: Poor: The experimental evaluation is flawed or the results fail to adequately support the main claims.

**Q2(5) Reproducibility:**

2: Fair: Key resources (e.g., proofs, code, data) are unavailable but key details (e.g., proof sketches, experimental setup) are sufficiently well-described for an expert to confidently reproduce the main results.

**Q3 Main Strengths:**

1. A theoretical analysis could be useful to understand the properties of the proposed method.

2. A sparse subspace clustering has been proposed. The proposed method looks simple to implement.


**Q4 Main Weakness:**

1. The proposed method has been compared to methods that are proposed more than 5 years ago.  It is hard to tell how the proposed method is compared with the state of the art.

2. The presentation of the paper may need substantial improvement.

3. It is not clear how the proposed method is related to recently proposed sparse subspace clustering methods.


**Q5 Detailed Comments To The Authors:**

There are two main concerns of the paper.  The first one is that although the proposed method was shown to have superior performance on reasonable collection of data sets, the methods used for comparison are not recent.  Those methods are proposed in or earlier than 2015.  The experiment includes also the l^0-SSC proposed in 2016.  However, it seemed that the results of it were not included in Table 1.  Besides, it was said that the proposed method has the same performance as l^0-SSC.  Hence, the experimental results could not sufficiently demonstrate the effectiveness of the proposed method.

The second concern is on the presentation.  The paper may need to be better organised.  Current, the discussion of the different methods in Section 1 may not be easier to understand without some review on background.  Moreover, the technical content may need to add more intuitive explanation to help readers understand.  Furthermore, some parts of the paper sometimes refer to later sections.  In section 1.1, there seems to have quite some repetition in explaining the contribution.  The sudden mathematical description in section 1.1 also looks a bit out of place.  These issues could possibly be addressed by better organisation of the content.

A less major concern is that the paper has not discussed the relationship of the proposed method with other more recent subspace clustering methods.  The latest subspace clustering methods that have been reviewed appears to be one proposed in 2018.

**Q7 Justification For Your Score:**

The paper provides some potential useful theoretical analysis and a novel method for subspace clustering. However, the effectiveness of the proposed method has not been verified empirical and the presentation of the paper needs to be improved.

**Q9 Complying With Reviewing Instructions:**

1: Yes.

---

### Decision · Program_Chairs · 2022-05-15

**Decision:**

Accept (Poster)

**Comment:**

Meta Review: The authors have done a remarkable job at addressing the reviewer concerns and the feedback great improves the clarify and significance of this work.  It is therefore imperative that the authors revise their manuscript as they proposed to do in the summary of their responses. Such revision will certainly prevent any misunderstandings on the paper contributions and make a clear case in favor of the proposed approach.